# Validation of an Eastern Armenian breast cancer health belief survey

**Haley Tupper**[1]*, **Razmik Ghukasyan**[1], **Armine Bayburtyan**[2], **Arin Balalian**[3], **Arsine Kolanjian**[4], **Marine Hovhanissyan**[2], **Shant Shekherdimian**[5]

**1** Department of General Surgery, University of California, Los Angeles, Los Angeles, California, United States of America, **2** School of Public Health, Yerevan State Medical University, Yerevan, Armenia, **3** Department of Epidemiology, Mailman School of Public Health, Columbia University, New York, New York, United States of America, **4** Department of Neuroscience, University of California, Berkley, Berkley, California, United States of America, **5** Department of Pediatric Surgery, University of California, Los Angeles, Los Angeles, California, United States of America

☯ These authors contributed equally to this work.

* htupper@mednet.ucla.edu

**Data Availability Statement:** The de-identified data is available in Supplemental S1 File.

**Funding:** The authors received no specific funding for this work.

## Abstract

With the fourth highest breast cancer mortality rate in the world, breast cancer prevention and early detection is a priority for Armenia. The Ministry of Health recently initiated efforts to expand access to breast cancer screening. However, little is known about the population's understanding and perception of breast cancer screening programs. This cross-sectional telephone-based study sought to develop and validate an Eastern Armenian language version of the Champion's Health Belief Model Scale (CHBMS) for future use. The English-language CHBMS survey was first rigorously translated by two Armenian nationals and evaluated for face validity. Telephone surveys were then administered to randomly-selected women of approximately screening age (35–65 years) with no prior history of breast cancer living in Armenia's capital between 2019–2020 (n = 103). The translated survey's psychometric properties were evaluated, examining (1) content equivalence, (2) test-retest reliability and (3) internal consistency. Content equivalence and test-retest reliability of the Armenian CHBMS were characterized using correlational analysis with Pearson's coefficient ranging from 0.76–0.97 (p<0.001) and 0.72–0.97 (p<0.001), respectively, for all five CHBMS domains. The translated survey's internal consistency was comparable to the original English-language CHBMS with a Cronbach's alpha greater than 0.7 for all five domains (0.75–0.94 (p<0.001). The translated Eastern Armenian version of CHBMS is a valid, internally-consistent, and reliable research tool that is ready for imminent use among screening-age women to investigate breast cancer perceptions and beliefs as the Armenian government seeks to expand screening access.

## Introduction

Breast cancer is an important cause of mortality among adult women in Armenia, an upper-middle income country in the South Caucasus. Statistics are particularly concerning among Armenian women ages 15–49: In this age group, breast cancer proportionally causes nearly

**Competing interests:** The authors have declared that no competing interests exist.

three times as many deaths as worldwide (14% vs. 5% of deaths), with a mortality to incidence ratio of nearly 50% [1, 2]. Many of these deaths are preventable by addressing risk factors, screening and improving treatment. Specifically, meta-analyses indicate that screening via mammogram reduces breast cancer mortality by 15–20% [3, 4]. Many higher-income nations have implemented successful breast cancer screening strategies. However, despite the lethality of breast cancer in Armenia and the proven efficacy of breast cancer screening in reducing mortality, Armenia does not currently have an organized breast cancer screening strategy [5].

The Republic of Armenia's Ministry of Health (MoH) is working on implementing an organized breast cancer screening program. To ensure widespread acceptance and successful implementation of the program, the MoH is exploring the public's awareness and perceptions of breast cancer and screening. Individual health behaviors, such as screening, are strongly influenced by individual health beliefs [6]. One of the most commonly-utilized and widely-validated models to evaluate individual health beliefs influencing breast cancer screening behavior is the Champion's Health Belief Model Scale (CHBMS) [6]. The CHBMS integrates several theories of health behavior and has been adapted and validated for populations across the globe [7–21]. However, no validated versions of the CHBMS existed in Eastern Armenian, the national dialect of Armenia, prior to this study. Given validation of a translated survey instrument prior to widespread utilization is a best practice, the purpose of this study was to evaluate the psychometric properties of the Eastern Armenian version of the CHBMS for imminent use.

## Methods

This cross-sectional telephone-based study collected basic demographic and breast cancer screening information and evaluated the psychometric properties of the translated CHBMS survey instrument. Institutional review board (IRB) exemption was obtained from the Office of Human Research Protection Program at the University of California, Los Angeles (#19–001507). Local institutional review board approval was also obtained from the Ethics Committee at Yerevan State Medical University (N˚3-2/19).

### Survey design

The revised version of the CHBMS was selected as the survey instrument for this study [6]. The CHBMS has five domains: 1) perceived susceptibility, 2) perceived benefits, 3) perceived barriers, 4) self-efficacy, and 5) fear. In addition to the CHBMS questions, the survey included questions exploring participants' familiarity with breast cancer, screening awareness, and willingness to pay for mammography, as well as several demographic questions, reported below in Table 1. The questionnaire had a total of 83 questions, 47 of which were CHBMS questions. The survey was translated from English to Armenian and back-translated into English independently by two translators to ensure accuracy. The survey was then refined and evaluated for face validity with local Armenians. Trained local surveyors screened participants in accordance with eligibility guidelines and obtained informed verbal consent, which was documented as part of the data collection. Survey responses were collected using the Qualtrics data collection software.

### Study population & data collection

A stratified random sampling approach was used to recruit women ages 35–65 proportionally from each of Yerevan's 12 administrative districts between 2019–2020. The age criteria was determined by the MoH's proposed breast cancer screening target group. Based on their address, registered residents of Yerevan are assigned to a public district polyclinic, which

**Table 1. Baseline characteristics of the study cohort.**

|  | n (%) |
|---|---|
| Total Respondents | n = 103 |
| Age (Median, IQR) | 47 (15.5) |
| Insurance Status |  |
| Insured | 26 (25.2%) |
| Uninsured | 77 (74.8%) |
| Marital Status |  |
| Married | 73 (70.9%) |
| Single | 16 (15.5%) |
| Divorced | 6 (5.8%) |
| Widowed | 8 (7.8%) |
| Highest Education Completed |  |
| High School | 22 (21.6%) |
| Vocational Degree | 28 (27.5%) |
| College Graduate and Beyond | 52 (51.0%) |
| Employment Status |  |
| Full-time Employed | 55 (53.9%) |
| Part-time Employed | 12 (11.8%) |
| Unemployed | 35 (34.3%) |
| Estimated Monthly Expenses |  |
| <100,000 AMD (<$200) | 20 (19.8%) |
| 100–300,000 AMD ($200–600) | 65 (64.4%) |
| 300–500,000 AMD ($600–1,000) | 13 (12.8%) |
| >500,000 AMD (>$1,000) | 3 (3.0%) |
| Overall Health Status |  |
| Excellent | 29 (28.2%) |
| Good | 70 (68.0%) |
| Poor | 4 (3.9%) |
| Family History |  |
| Breast Cancer | 23 (22.3%) |
| Breast Cancer Mortality | 11 (10.7%) |
| Breast Cancer Screening History |  |
| Self-Exam | 55 (53.4%) |
| Mammography | 19 (18.4%) |
| Ultrasound | 14 (13.6%) |
| Magnetic Resonance Imaging | 1 (1.0%) |
| Multiple Imaging Modalities | 6 (5.8%) |

provides outpatient primary and specialty care, and the MoH maintains a list of all polyclinic registrants, including their phone numbers. Study investigators obtained the phone numbers of all female residents of Yerevan ages 35–65, stratified by district, from the MoH. Potential study participants were then proportionally selected from each district using a random number generator to receive a telephone call to undergo the survey. The district stratification was maintained because each district represents unique socioeconomic and healthcare access realities. Phone calls were made after 5pm to account for work hours and each number was attempted once. Women with a prior breast cancer diagnosis or surgery for breast mass were excluded. Two unique minimum sample size calculations were performed given collection of two distinct data types, specifically cross-sectional survey data and translated survey

psychometric property evaluation data. For the cross-sectional survey portion (Table 1), a minimum sample size of 101 respondents was calculated to be representative of the study population (confidence level: 95%, margin of error: 5%), given women ages 35–65 residing in Yerevan represented approximately 7% of the Armenian population of 2.9 million in 2019 [22]. For the evaluation of an instrument's psychometric properties, there is no consensus regarding sample size specifications, but a simple flat minimum of 100 participants has been found to be adequate [23]. A total of 759 women were called and 185 consented to participate (24.4%); 43 of those were excluded based on age and/or prior breast cancer diagnosis and another 39 were not included in the analysis due incomplete data entry. Data entry was considered complete if more than 90% of the CHBMS questions were answered. Ultimately, 103 completed surveys were included in the analysis.

To ensure content equivalence to the original English version, the survey was administered twice (once in English and once in Armenian) to 20 additional participants fluent in both languages with a 10–14 day latent time between surveys. These participants were women ages 35–65 residing in the United States with no personal history of breast cancer or breast biopsies identified by convenience sampling. Responses in each language were then correlated. To evaluate the reliability of the survey over time, or test-retest reliability, the Armenian survey was also re-administered to 24 of the 103 participants, again with a 10–14 day latent time, and responses over time were correlated.

## Statistical analysis

The data was analyzed using IBM SPSS Statistics 27. The statistical analysis included 1) Descriptive statistics of demographic data, 2) Correlational analysis, reporting on Pearson's correlation, and 3) Internal consistency analysis, reporting on Cronbach's alpha coefficient. Test-retest reliability and content equivalence of the Armenian CHBMS were both evaluated by correlational analysis. Cronbach's alpha of 0.7 or greater was indicative of acceptable internal consistency.

## Results

### Demographics

A total of 103 telephone surveys were completed and analyzed. Full demographic data is available in Table 1. The median age of participants was 47. Most women were married (70.9%), spent 100,000–300,000 Armenian Dram (AMD) per month (64.4%), and had completed either a vocational or college degree (78.5%). These women were variably employed (53.9% full-time employed, 11.8% part-time employed and 34.3% unemployed) and largely uninsured (74.8%). Almost one-quarter of respondents (22.3%) had a family history of breast cancer (1st or 2nd degree relative) and over one in ten respondents had a family member who had died from breast cancer. While over half of respondents had done breast self-exam in the past, only 18.4% had previously undergone a mammogram.

### Survey validation

The translated version of the survey was found to have good content equivalence between languages with a Pearson's R coefficient ranging from 0.76–0.97 (p<0.001) for all 5 CHBMS domains (susceptibility, benefits, barriers, self-efficacy, fear) (see Table 2). Similarly, the survey demonstrated strong test-retest reliability across all 5 CHBMS domains with Pearson's R coefficient ranging from 0.72–0.97 (p<0.001) (see Table 3). Lastly, the Armenian CHBMS demonstrated acceptable internal consistency with a Cronbach's alpha coefficient ranging from 0.75–

**Table 2. Content equivalence of translated survey.**

|  | Pearson's R Coefficient | p-value |
|---|---|---|
| Domain |  |  |
| **Susceptibility** (n = 20) | 0.76 | <0.001 |
| **Benefits** (n = 20) | 0.83 | <0.001 |
| **Barriers** (n = 20) | 0.96 | <0.001 |
| **Self-Efficacy** (n = 20) | 0.97 | <0.001 |
| **Fear** (n = 20) | 0.92 | <0.001 |

0.94 in all five domains of CHBMS (see Table 4). These values of internal consistency amongst domains were comparable to the values obtained in the original CHBMS study [6].

## Discussion

Armenia does not have an organized breast cancer screening program [5]. To date, breast cancer screening is not part of the basic benefits package, although mammograms are covered under some social packages with employers [24]. Mammogram coverage is very limited: In a 2016 health system assessment, among women ages 30–60, 8.8% had undergone a mammogram in the last three years, 6.9% had undergone a mammogram more than three years ago, and 84.4% had never had a mammogram [24]. Of the women who had undergone a mammogram in the last three years, 59% had paid out of pocket for the test, 19% had used a social package and the remaining 22% had been covered by some combination of free exams, insurance and other sources [24]. The high number of respondents who were uninsured in this study (74.8%) is representative of healthcare realities in Armenia, where 62% of Armenians must pay entirely out of pocket for healthcare services not included in the basic benefits package [25]. In the 2016 assessment, women with higher education and incomes had undergone proportionally more mammograms [24]. Low screening rates result in delayed detection with falsely low incidence rates. Consequently, in the setting of largely opportunistic diagnosis, Armenia has the highest proportion of breast cancer cases diagnosed at Stage IV (20.8%) and the highest mortality-to-incidence ratio among former Soviet states (excludes Baltic states) [2]. Compared to global breast cancer mortality rates, Armenia's breast cancer mortality rate is 41% higher without a clear etiology [26, 27].

The Armenian Ministry of Health is in the process of implementing a universally-covered organized breast cancer screening protocol. While individual cost for screening is an obvious barrier, it is not the only one. Studies have repeatedly demonstrated that individual health beliefs strongly influence health behavior uptake, a key factor in screening success [6]. Champion's Health Belief Model Scale has been instrumental in assessing beliefs that impact screening behavior across the globe, and has been translated and adapted to more than a dozen different cultural contexts with strong reliability and validity [7–21]. However, for each

**Table 3. Test-retest reliability of translated survey.**

|  | Pearson's R Coefficient | p-value |
|---|---|---|
| Domain |  |  |
| **Susceptibility** (n = 24) | 0.72 | <0.001 |
| **Benefits** (n = 24) | 0.78 | 0.001 |
| **Barriers** (n = 24) | 0.97 | <0.001 |
| **Self-Efficacy** (n = 24) | 0.86 | <0.001 |
| **Fear** (n = 24) | 0.85 | <0.001 |

**Table 4. Internal consistency of translated survey.**

|  | Cronbach's α Coefficient | Survey Items (N) |
|---|---|---|
| Domain |  |  |
| **Susceptibility** (n = 102) | 0.94 | 4 |
| **Benefits** (n = 102) | 0.87 | 4 |
| **Barriers** (n = 99) | 0.92 | 24 |
| **Self-Efficacy** (n = 102) | 0.75 | 9 |
| **Fear** (n = 102) | 0.91 | 6 |

language and culture, this validation process–the examination of the survey instrument's psychometric properties—must be repeated prior to widespread use. To date, no survey instrument had been validated in Armenian to assess health beliefs regarding breast cancer.

The Armenian CHMBS has been found to be reliable and valid among Armenian women and is ready for imminent use. Not only does the Eastern Armenian version subjectively continue to assess breast cancer screening health beliefs (face validity), but survey content is objectively equivalent in English and Armenian (content equivalence) with consistent responses over time (test-retest reliability). Furthermore, individual items purported to measure the five constructs of perceived susceptibility, perceived benefits, perceived barriers, self-efficacy, and fear continue to reliably measure these constructs (internal consistency) with comparable results to the original English CHBMS survey.

## Potential limitations

The main potential limitations to this study include incomplete answers and possible unmeasured differences between those who agreed and those who declined to participate. Approximately one-quarter of responses were excluded due to incompleteness (39 of 142 responses). Survey external validity may also be limited by the surveyed population; anecdotally, Yerevan's population is generally more educated with associated increased health literacy. Furthermore, while Eastern Armenian is the national dialect, there has been a recent increase in immigration with an influx of Western Armenian speakers.

## Next steps

With the fourth highest breast cancer mortality rate in the world, breast cancer screening is a priority for Armenia [28]. Now that the survey has been refined and validated, it will be administered Armenian women to better understand health beliefs that would influence screening behaviors. Characterizing women's health beliefs towards screening will facilitate public health communication strategies as national breast cancer screening is rolled out. Anecdotally, local providers, lay Armenians and international visitors alike have all observed that breast cancer affects Armenians at a younger age with a tendency to be more aggressive. Further study of this observation is necessary with increased phenotyping of diagnosed breast cancers to inform both screening strategies and treatment. Given Armenian physicians' historic facility with ultrasound and the potential need to screen younger patients with denser breast tissue, the role of ultrasound for screening in Armenia also requires further evaluation. Finally, quality improvement of the continuum of diagnosis to treatment, specifically improved histopathologic diagnosis and biopsy technique to increased availability of necessary chemotherapeutic medications, is vital as screening strategies are rolled out.

## Conclusion

Given the burden of breast cancer among Armenian women, further research into screening strategies, attitudes and public health messaging is needed urgently. The translated Armenian version of CHBMS is a valid, internally-consistent, and reliable research tool that can be utilized to investigate current beliefs and perceptions about breast cancer among Eastern Armenian speaking women as the Armenian Ministry of Health begins to design and implement its national breast cancer screening strategy.

## Supporting information

**S1 File. De-identified data.**
(XLSX)

**S2 File. Armenian language survey with CHBMS.**
(DOCX)

**S3 File. English language version survey with CHBMS.**
(DOCX)

## Acknowledgments

The study authors would like to acknowledge Mary Mkhitaryan for her diligent assistance with data collection and Kim Hekimian for her advice on the study design.

## Author Contributions

**Conceptualization:** Razmik Ghukasyan, Arin Balalian, Arsine Kolanjian, Marine Hovhanissyan, Shant Shekherdimian.

**Data curation:** Razmik Ghukasyan, Armine Bayburtyan, Arsine Kolanjian.

**Formal analysis:** Haley Tupper, Razmik Ghukasyan.

**Investigation:** Razmik Ghukasyan.

**Methodology:** Razmik Ghukasyan, Shant Shekherdimian.

**Project administration:** Razmik Ghukasyan, Shant Shekherdimian.

**Resources:** Shant Shekherdimian.

**Software:** Razmik Ghukasyan.

**Supervision:** Marine Hovhanissyan, Shant Shekherdimian.

**Validation:** Haley Tupper, Arin Balalian.

**Visualization:** Haley Tupper, Razmik Ghukasyan.

**Writing – original draft:** Haley Tupper, Arsine Kolanjian.

**Writing – review & editing:** Haley Tupper, Shant Shekherdimian.

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
