## [Decision Letter · Decision Letter 0]

3 Feb 2023

PGPH-D-22-02079

Validation of an Eastern Armenian breast cancer health belief survey

Dear Dr. Tupper,

Thank you for submitting your manuscript to PLOS Global Public Health. After careful consideration, we feel that it has merit but does not fully meet PLOS Global Public Health’s publication criteria as it currently stands. Therefore, we invite you to submit a revised version of the manuscript that addresses the points raised during the review process.

We look forward to receiving your revised manuscript.

Kind regards,

Nnodimele Onuigbo Atulomah, PhD

Academic Editor

Journal Requirements:

3. Please amend your Data Availability Statement and indicate where the data may be found.

Additional Editor Comments (if provided):

The three reviewers were clear about the appropriateness of the study but pointed out a few revisions that would strengthen the manuscript. Kindly pay keen attention to every details pointed out for revision.

Reviewers' comments:

Reviewer's Responses to Questions

**Comments to the Author**

1. Does this manuscript meet PLOS Global Public Health’s publication criteria? Is the manuscript technically sound, and do the data support the conclusions? The manuscript must describe methodologically and ethically rigorous research with conclusions that are appropriately drawn based on the data presented.

Reviewer #1: Yes

Reviewer #2: Yes

Reviewer #3: Yes

2. Has the statistical analysis been performed appropriately and rigorously?

Reviewer #1: Yes

Reviewer #2: Yes

Reviewer #3: Yes

3. Have the authors made all data underlying the findings in their manuscript fully available (please refer to the Data Availability Statement at the start of the manuscript PDF file)?

Reviewer #1: Yes

Reviewer #2: No

Reviewer #3: Yes

4. Is the manuscript presented in an intelligible fashion and written in standard English?

Reviewer #1: Yes

Reviewer #2: Yes

Reviewer #3: Yes

5. Review Comments to the Author

Reviewer #1: This manuscript presents an evaluation of the psychometric properties of Eastern Armenian version of the Champion’s Health Belief Model Scale (CHBMS). The manuscript is generally well written and clearly presented.

Other general and specific comments are provided below:

1. Originality of value: The manuscript presents an original study which was designed and conducted among women age 35-65years living in Armenian Capital. The paper addressed a relevant public health issue of concern, namely breast cancer screening with results to prove the validity of the instrument.

2. Suitability and soundness of technique: The techniques used for the study are adequate and robust.

3. Clarity of Presentation: The manuscript was remarkably clear in narrative presentation. It is easy to follow and complied with the journal specifications.

4. Areas requiring correction. The paper is generally well written, however, there are few areas I will like the authors to correct;

On the method line 79; the ethical review should be place after the study design.

Line 83: the type of study design was not mentioned likewise sample size was determination.

On the conclusion line 230, the author stated that” the research tool can be utilized to investigate current attitude’’ I do not agree with the assertion. The instrument can only measure perception and belief. The author could remove the attitude from that statement.

Reviewer #2: 1. The author may wish to declare the research Ethic clearance number which is usually given by the review Board.

2. For evidence based approach, it will not be out of context if the researchers display the English version of the items contained in the five constructs; and also the translated version which was done into Eastern Armenian Language. . A-38 item construct is huge and will be better to mention just a few if not all.

3. The authors need to subject this manuscript to grammar checks and accuracy. Couple of grammatical errors /omission of conjunctions were obvious.

4. There was no clarity on how the researchers got the phone numbers of the participants. I suggest this should stated clearly on the methodology. Were there any encounters during this process of telephoning? No statement captured this.

5. The authors to make available the translated version for future use by other researchers in the study area.

Reviewer #3: 1. The manuscripts meets the standard of PLOS Global Public Health

2. The statistical analysis are in order

3. The authors are willing to make all their data publicly available without restriction. They have agreed to the journal Data Policy.

4. The manuscripts has been written in standard English language.

Other Issues for revision by the authors

• How was the random sampling done? Is it according to districts or street numbers of based on clusters within the state capital? There is need for more clarity on this statement. See 35-36 on Page 2.

• The authors should consider moving the statement on Institutional review board (IRB) on Line 79 to Line 131”. Page 4.

• The authors need to provide complete details of the exemption obtained from University of California Office of Human Research Protection Program on Line 80 Oage 4.

• The authors need to provide the Ethical Approval Number of the study. That is details of the ethics approval that was obtained from Yerevan State Medical University. Page 4.

• Why the need for stratified random approach? The authors need to provide more details. See 96 on Page 4

• The Table 1 on Page 5 reports that 79(76.7%) of the respondents were uninsured and 35(34.0%) were unemployed. Based on realities in Armenian are these likely to affect the way the respondents have responded to the questions.

• The authors need to provide details and results on the 24 additional participants that are both fluent in English Language and Armenian Language. See 127 on Page 6.

• The authors need to correct the IBM SPSS statistics version 7 to the most recent version of the software. Is this the correct SPSS version used for this study’s data analysis. See 134 on Page 6.

6. PLOS authors have the option to publish the peer review history of their article (what does this mean?). If published, this will include your full peer review and any attached files.

**Do you want your identity to be public for this peer review?** For information about this choice, including consent withdrawal, please see our Privacy Policy.

Reviewer #1: **Yes: **Titilayo Olaoye

Reviewer #2: **Yes: **Khadijat Toyin Musah

Reviewer #3: **Yes: **Saheed Akinmayowa Lawal

---

## [Editor Report · Decision Letter 1]

13 Apr 2023

Validation of an Eastern Armenian breast cancer health belief survey

PGPH-D-22-02079R1

Dear Dr. Tupper,

We are pleased to inform you that your manuscript 'Validation of an Eastern Armenian breast cancer health belief survey' has been provisionally accepted for publication in PLOS Global Public Health.

Best regards,

Nnodimele Onuigbo Atulomah, PhD

Academic Editor

The suggested revisions have been satisfactorily completed. With no other revision emerging and required this manuscript is ready for the next stage of publication.